| **Open Peer Review** | Clinical Microbiology | Methods and Protocols

# Empowering global disease surveillance with CURED: a tool for rapid identification of unique genomic biomarkers

Erin Theiller,[1,2] Swetha Rajagopol,[3] Stephanie Higgins,[1,2] Dayanara I. Torres,[1,2] T'Nia Napper,[1,2] Bianca E. Galis,[1,2] Arie Dash,[1,2] Elizabeth Qian,[2,4] Lauren Hamlette,[2,5] Qianxuan She,[1,6] Ceylan Tanes,[1,2] Nathan L'Etoile,[3,6] Andries Feder,[3] Alice Slotfeldt Viana,[7] Matheus Assis Côrtes Esteves,[7] Michael C. Abt,[8,9] Susan E. Coffin,[3,6] Ericka Hayes,[3,6] Robert F. Potter,[1,10,11] Joseph P. Zackular,[1,10,11,12] Lakshmi Srinivasan,[6,13] Agnes Marie Sá Figueiredo,[7,14] Paul J. Planet,[1,3,6,15] Ahmed M. Moustafa[1,2,6,16]

**ABSTRACT**  Rapid tracking of emerging pathogenic microorganisms is crucial for designing effective treatment, infection control, and prevention strategies. While whole-genome sequencing (WGS) offers the necessary granularity to track emerging clones, it remains prohibitively expensive at the scales needed to monitor with high resolution in real time. We present CURED (Classification Using Restriction Enzyme Diagnostics), which uses a training set of sequenced genomes to identify unique k-mers with restriction sites specific to a clonal lineage. CURED enables fast and inexpensive PCR-based diagnostic tests for surveillance or outbreak investigations with minimal use of WGS. Benchmarking against existing tools, CURED compares favorably and scales more efficiently than other k-mer search strategies. We validated and tested CURED in five distinct data sets: (i) previously identified biomarkers described for a methicillin-resistant *Staphylococcus aureus* (MRSA) clone in Rio de Janeiro, (ii) diagnostic alleles for different lineages in the USA300 MRSA clone, (ii) the extensively drug-resistant *Acinetobacter baumannii* Global Clone 1 lineage, (iv) toxigenic versus non-toxigenic *Clostridioides difficile*, and (v) circulating *S. aureus* clones in a neonatal intensive care unit (NICU). We implemented CURED as part of NICU infection prevention efforts and report the test's speed, sensitivity, and specificity in a real-world setting. CURED is a scalable, multithreaded, memory-, and cost-efficient pipeline tailored for rapid clone detection and restriction site analysis. While particularly impactful for localized outbreak investigations and targeted surveillance, our preliminary work at the global scale suggests broader implementation is feasible. CURED is freely available at https://github.com/microbialARC/CURED.

**IMPORTANCE** Timely and cost-effective detection of emerging microbial clones is essential for infection prevention and public health surveillance. While whole-genome sequencing remains the gold standard for tracking microbial evolution and transmission, its cost, infrastructure requirements, and turnaround time limit its scalability, especially in resource-limited settings. CURED addresses this gap by enabling the development of inexpensive, PCR-based diagnostic assays informed by genomic data, without requiring further sequencing. By identifying lineage-specific restriction sites through a scalable and memory-efficient k-mer pipeline, CURED enables the translation of genome-scale insights into actionable diagnostics. This tool supports broader implementation of genomic-informed diagnostics in both local and global pathogen surveillance efforts.

**KEYWORDS**  clonal lineage tracking, rapid pathogen surveillance, microbial diagnostics, k-mer analysis, restriction enzyme PCR, hospital, infection prevention and control, outbreak, resource-limited settings

**Peer Reviewer** Jonathan C. Thomas, Nottingham Trent University, Nottingham, United Kingdom

Address correspondence to Ahmed M. Moustafa, moustafaam@chop.edu, Paul J. Planet, planetp@chop.edu, or Agnes Marie Sá Figueiredo, agnes@micro.ufrj.br.

The authors declare no conflict of interest.

See the funding table on p. 13.

Tracking emerging hypervirulent and adapted pathogenic viruses, bacteria, and fungi is an imperative for public health. While whole-genome sequencing (WGS) has become the gold standard for surveying such pathogens, access to resources and the associated costs make applying WGS for larger numbers of isolates prohibitive in resource-rich settings, and such surveillance is nearly impossible in resource-scarce settings. Consequently, there is a growing need for bioinformatic tools that can leverage WGS data to quickly find diagnostic, apomorphic sequence mutations that are unique to the clone of interest. These identified biomarkers can then be used to create a diagnostic screen. While existing tools such as KmerGO and KEC offer solutions for identifying unique k-mers and designing diagnostic assays, their computational demands (memory and time) can be prohibitive, and they do not provide a way to specify or enforce explicit sensitivity and specificity thresholds during biomarker selection (1, 2).

Here, we introduce the Classification Using Restriction Enzymes Diagnostics (CURED) pipeline, which uses local and public sequencing data, integrating rapid k-mer analysis and restriction enzyme (RE)-based detection in a single pipeline. CURED is capable of processing tens of thousands of genomes within a few hours. The goal of the algorithm is to identify highly specific and sensitive k-mers with exclusive restriction sites, enabling the design of PCR-based assays that rely on REs rather than sequencing of the product. The output of the pipeline is a genomic region containing the unique k-mer with unique RE to be used for primer design for PCR amplification. After amplification, restriction endonuclease digestion results in a unique gel electrophoresis pattern for the clone of interest. To gauge its utility, we tested CURED in five distinct genomic data sets, from representative Gram-positive, Gram-negative, and anaerobic pathogens, ranging from local to global, and observed diagnostic k-mer/RE sites in all of them.

## RESULTS AND DISCUSSION

### Implementation

The CURED pipeline is composed of two scripts developed in Python and tested on Linux and macOS machines. The first step of the pipeline runs CURED_Main.py, which finds k-mers unique to the genomes in a "case" group compared to a "control" group. CURED_FindREs.py is run in the second part of the pipeline to find k-mers in this k-mer subset with restriction enzyme sites that are present only in the case group. These k-mers are referred to as biomarkers in the context of this study. The designation of genomes into case and control groups in CURED is flexible and user-driven. Users can define groups based on phylogenomic clustering, sequence type (ST), other molecular typing schemes, or phenotypic characteristics of interest. Case groups are typically selected from a specific lineage or molecular cluster, while control groups can be assembled using the user's own curated data sets or by leveraging publicly available genome collections.

As illustrated in Fig. 1, there are many entry points to the CURED pipeline, allowing flexibility based on the user's available data and study design:

i. Providing local sequencing data: users have the option to provide local sequencing data, designating the genomes as either case or control based on the groupings they have curated.
ii. Supplementing custom data sets with public genomes: users can provide sequencing data for a specific set of genomes under investigation and supplement it with publicly available genomic data from NCBI as controls. In this scenario, the user must identify their species of interest, and CURED uses public data sets to download the corresponding genomes from the NCBI database (3).
iii. Specifying a species and ST: users can also specify a species and a particular ST of interest to be analyzed as the case set. When this option is selected, after downloading the genomes corresponding to this species from the public database, CURED uses MLST (v 2.23.0) to carry out multi-locus sequence typing

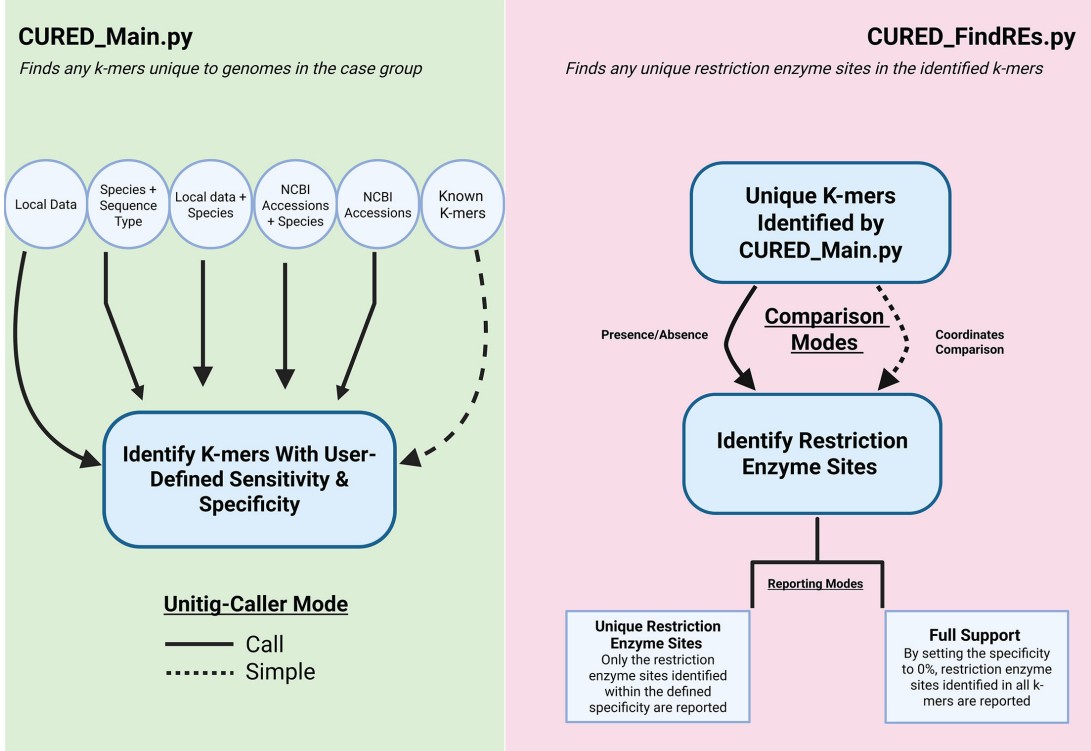

**FIG 1** Overview of the CURED pipeline. A schematic of the CURED workflow. In stage 1, CURED identifies k-mers specific to a user-defined group of genomes. In stage 2, the pipeline detects restriction enzyme recognition sites unique to the selected k-mers.

(4, 5). Genomes belonging to the designated ST form the case group, while genomes not belonging to this ST serve as the control group. In contrast, when users provide their own case and control genomes based on non-ST criteria (e.g., phenotype, outbreak association, or clinical outcome) as in entry point one, MLST is not performed, and group assignment is fully user-defined.

iv. Providing a text file of NCBI accessions: users have the flexibility to provide a text file of NCBI accessions to CURED. This file may list only case accessions (along with a species designation) or include both case and control accessions for a more targeted analysis.

v. Using a predefined k-mer set: users may supply a known set of k-mers to query against the case and control genomes defined by any of the previous methods.

## Efficient unique k-mer discovery using iterative analysis

Under the hood, CURED leverages multiple iterations of unitig-caller (v 1.3.0), which is a wrapper around Bifrost, a program for parallel construction, indexing, and querying of colored and compacted de Bruijn graphs (6). By utilizing a combination of the multiple modes of unitig-caller, as well as breaking down the data set into smaller batches and making use of multiple iterations, the CURED pipeline operates in a fast, memory-efficient manner. Briefly, as the aim is to identify unique sequences within a targeted set of genomes, the user-defined sensitivity threshold is analyzed first. As such, in the first iteration of unitig-caller, most or all of the case-designated genomes are included in the search space, along with only a few control-designated genomes. This procedure allows the algorithm to identify k-mers that are found in all the case genomes but in none of the controls. This sensitivity-first, batching method enables the analysis to be conducted by effectively decreasing both run time and memory usage. In the subsequent iterations, which are defined by any remaining case genomes and a set number of control genomes, CURED identifies if the already-found k-mers are present or

absent in these genomes. After each iteration, the identified unitigs are parsed according to the user-set sensitivity and specificity thresholds, carrying over any unitigs into future iterations if they are within the defined limits.

To further increase the efficiency of the tool, different modes of unitig-caller are applied. The default mode is "call" mode, in which unitigs are extracted by constructing a population graph returning the colors associated with the unitigs. In this mode, unitig-caller also extends k-mers into the longest continuous unique sequence. As a result, consecutive k-mers that are unique to the case group are merged into a single contiguous region. This ensures that the flanking regions of the extended sequence are properly represented and reduce the likelihood that either flank would be absent from the control group. This mode is employed in the first iteration, while CURED leverages the "simple" mode option of unitig-caller in the subsequent iterations. With the exception of supplying CURED with a list of known k-mers, each entry point to the pipeline is run in this manner. When supplying the tool with a list of previously identified k-mers, the first iteration of unitig-caller is bypassed, and the tool goes directly into searching for the presence of these known k-mers in the genomes using simple mode. This multi-faceted approach ensures that CURED operates with high efficiency, balancing computational speed and memory management even when handling large genomic data sets.

## Restriction enzyme site characterization

Following the identification of any unique k-mers, these sequences are then checked to characterize potential restriction enzyme recognition sites specific to the case-designated group of genomes. To carry out this step, the unitig sequence is searched for in one case genome using either the BWA-MEM (v 0.7.17) or blastn (v 2.15.0) algorithm, depending on the length of the unitig (7–9). BWA-MEM is used first to try to find the unitig in the case genome. If BWA-MEM cannot find the sequence in the case genome due to its short length, blastn (with word size set to 10 and e-value set to 1e-2) is used. After identification, the sequence is analyzed for recognition sites using the Biopython restriction package (v 1.82) and extended by 20 bases (or a user-defined length) on both ends to allow for easier identification in the control genome (10). Following this step, this elongated sequence is then searched for in each of the control-designated genomes using the BWA-MEM algorithm. If found in a control genome, the sequence is extracted using samtools (v 1.18) and subsequently searched for the same recognition sites that were previously identified in the case genome (11).

The RE analysis portion of the pipeline can be run in two different modes. The default mode is based on the presence or absence of the RE recognition site to determine its uniqueness. For instance, if the same RE recognition site found in the case genome is in any portion of the sequence found in the control, that recognition site is considered not unique to the case group. The alternative mode for running this portion of the pipeline involves comparing the coordinates of the identified restriction enzyme site. If an RE site that is identified in the case genome is only found in the flanking region of the control sequence, then this particular restriction site is considered unique to the case group. In this context, the flanking region is defined as the region where the sequence was elongated. For example, an RE site is identified in a 20-mer that is unique to the case group. Once identified, the default number of bases, which is 20, is added to either end of the sequence, elongating the sequence to 60 bases in length. Upon identification of this longer sequence in a control genome, if the same RE that was identified in the case is identified in either the first 20 bases or the last 20 bases of the control sequence, then this particular RE site is still considered unique to the case group. This is because its position relative to the k-mer suggests that it would not generate the same banding pattern. Sequence coordinates are important in this context because they allow users to evaluate not just whether an RE site is present in the control group, but where it occurs, to ensure that a site is not excluded solely due to its presence in the control group, when, in fact, its genomic position may still support its uniqueness to the case group

due to a different gel banding pattern. In this same scenario, if the default mode was selected, then the mere presence of this particular RE site in the control, irrespective of the location, will render this restriction site not unique to the case. Thus, the different available options for searching for unique restriction enzyme sites allow the user to select a method that will best suit their needs.

## Customizable specificity and reporting options

In addition to different search modes, the RE analysis portion of the pipeline also allows for different reporting modes. The default reporting mode is 100% specificity, meaning that an RE recognition site is not detected in any of the controls, but the user has the ability to change this specificity threshold. Furthermore, setting the specificity to 0% results in the reporting of how often a particular recognition site appears in the control genomes. The modes of this portion of the pipeline can be mixed, meaning that the default mode of reporting a restriction enzyme site can be run with either a user-defined specificity or in full-support mode (i.e., setting specificity to 0%). The same flexibility holds true for the coordinate comparison mode. Regardless of the selected reporting mode, the tool also reports the names of the control genomes when a restriction enzyme site is identified, providing the user with the ability to further investigate a particular genome following the conclusion of the pipeline.

## Validation of the tool

We first validated CURED on a data set of *S. aureus* genomes from an emerging MRSA strain in Rio de Janeiro, Brazil. Previous genomic analysis determined that a 20-mer sequence containing a unique *Bgl*I endonuclease recognition site was highly specific and sensitive to this group of genomes, which is referred to as the "RdJ clone" (12, 13). Using the CURED pipeline, 154 RdJ genomes were compared to 79,776 non-RdJ *S. aureus* genomes (Table S1). CURED accurately identified the unique sequence previously reported (12, 13). The pipeline found the sequence with 100% sensitivity and 99.99% specificity, meaning that in addition to being present in all of the RdJ isolates, the sequence was also found in 12 genomes from other lineages, which is consistent with the results of the previous study (12, 13). Additionally, CURED correctly detected *Bgl*1 endonuclease as one of the unique restriction sites in this sequence (12, 13).

Following validation of CURED's accuracy using the well-characterized RdJ clone, we assessed its computational performance. Using the RdJ data set, we benchmarked CURED against two existing tools that can identify group-specific k-mers, KEC and KmerGO, focusing on runtime and memory usage (Fig. 2A). For each tool, we randomly selected subsets of the control group to pair with the case group and repeated the analysis three times per data set size to ensure reproducibility (Table S1). With default parameters, CURED achieved k-mer identification in a runtime comparable to KEC for the data sets with 10,000 genomes or more, but with more than an order of magnitude lower memory requirement—less than 3 GB of RAM versus over 135 GB for KEC. Further showcasing its efficiency, CURED successfully processed nearly 80,000 genomes in under 90 min when run with eight threads, consuming only 12 GB of RAM—still substantially less than KEC. Both CURED and KEC accurately identified the RdJ k-mer, validating CURED's results; however, CURED accomplished this with substantially lower resource requirements and faster runtimes (Table S1). KmerGO, another tool designed to identify group-specific k-mers, could not be run on the full data set due to its excessive runtime. It required over two days to analyze just 10,000 genomes and was therefore excluded from larger-scale comparisons (Fig. 2A). Even at this data set size, CURED significantly outperformed KmerGO in both runtime and memory usage. CURED is currently the only tool among those evaluated that supports multithreading and offers additional analytical features such as sensitivity, specificity, and restriction site analysis—capabilities not available in KEC or KmerGO. Notably, scanning the 79,754 non-RdJ genomes for unique RE recognition sites was completed in just 17 h using less than 1 GB of RAM. Although the restriction enzyme validation phase required roughly 17 h when

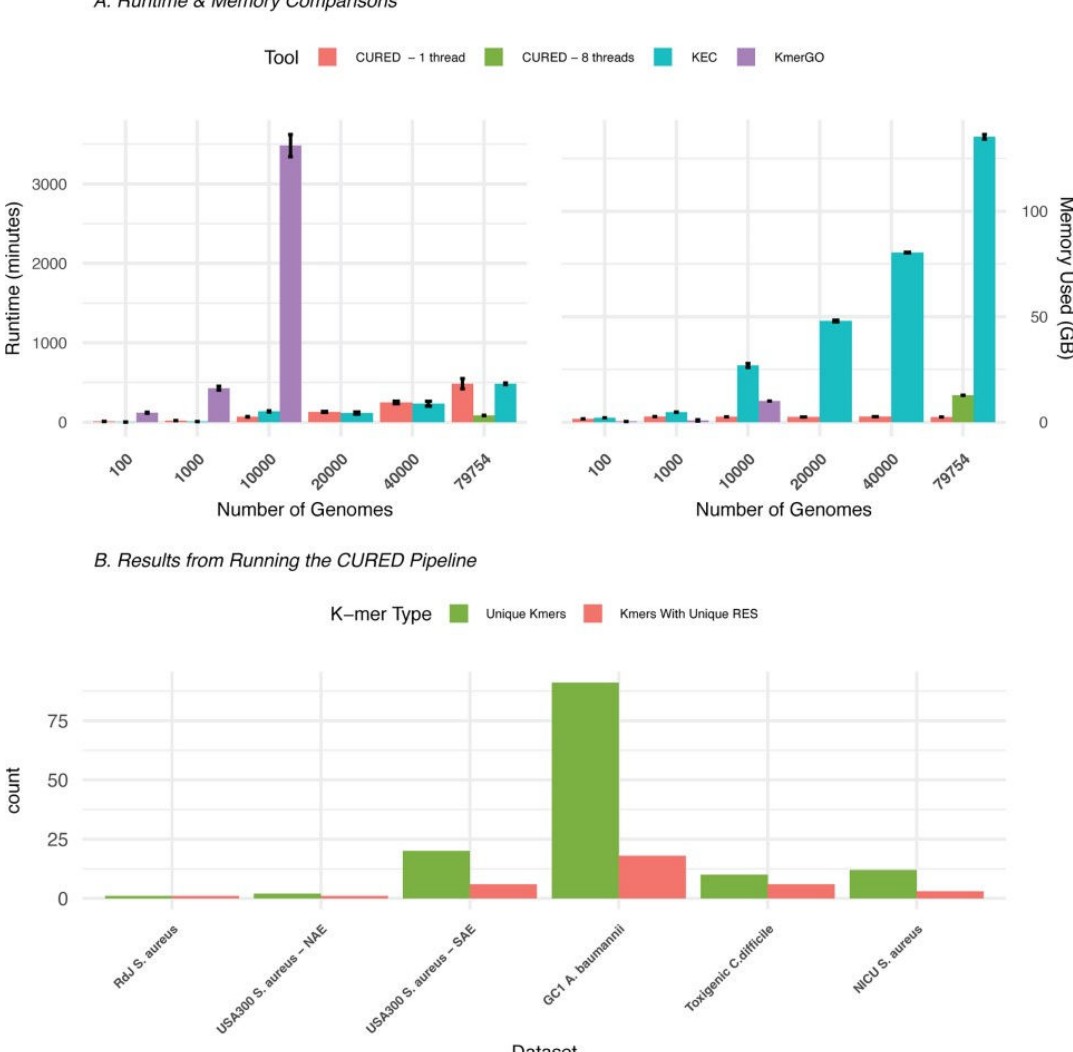

**FIG 2** Benchmarking of the CURED pipeline. (A) Benchmarking of CURED against existing k-mer-based tools across multiple data sets of increasing sizes. CURED was run using default parameters in single-threaded mode; on the largest data set, an additional run with eight threads demonstrates improved scalability and runtime. (B) Number of k-mers identified and percentage with restriction enzyme sites per data set.

applied to a very large global data set (>79,000 genomes), this step is performed only once per target lineage during assay development. Once a unique k-mer-restriction enzyme combination has been identified, the resulting assay can be deployed as a PCR diagnostic test, enabling same-day detection without the need for further sequencing. Overall, CURED demonstrated superior scalability, faster runtimes, and dramatically lower memory requirements compared to existing methods, particularly on large-scale data sets. These performance gains highlight CURED's suitability for real-time genomic surveillance and large-cohort comparative studies.

To assess CURED's versatility, we next evaluated its performance in the context of a more genetically diverse lineage. We applied CURED to a data set of global USA300 *S. aureus* genomes, a dominant community-associated MRSA clone characterized by substantial within-lineage diversity due to its international spread (14). In this case, CURED was used to validate an exact-matching approach for identifying diagnostic protein alleles, which were previously defined for each major USA300 clade (14). To evaluate CURED's ability to detect clade-specific biomarkers, the method was applied to the South American Epidemic (SAE) and North American Epidemic (NAE) clades, with a focus on determining whether any of the diagnostic alleles contained unique restriction

sites (Supplemental method). For the SAE clade, in under 10 min and requiring less than 1 GB of RAM, CURED identified 20 k-mers that were 100% sensitive and specific to this clade of 73 genomes. Of these 20 k-mers, 6 had unique restriction enzyme sites. Furthermore, CURED accurately identified the k-mers that overlapped with this clade's diagnostic alleles (Table S2) (14). Similarly, for the NAE clade of 129 genomes, CURED identified 2 k-mers with 100% sensitivity and specificity in under 10 min, using less than 1 GB of RAM; one of these k-mers had a unique restriction enzyme site. Similarly, CURED also accurately identified sequences contained within the diagnostic alleles of this clade (Table S2) (14). Notably, for the diagnostic protein allele PdhD, which was associated with the NAE clade, CURED identified a unique restriction site within the k-mer. In the original study, none of the protein-based diagnostic alleles characteristic of their respective clades achieved 100% sensitivity and specificity (14). In contrast, CURED identified unique sequences for the NAE and SAE clades with 100% sensitivity and specificity. In the case of the NAE clade, the distinguishing DNA-level changes were synonymous mutations that do not alter the amino acid sequence. The diagnostic allele approach failed to detect these sequences because it is limited to protein-level variation. Although CURED also identified unique k-mers for the SAE clade, these mapped to larger coding regions with additional variation, resulting in multiple protein alleles in the case group and preventing the identification of a single diagnostic protein allele with perfect sensitivity and specificity. These findings underscore CURED's advantage in detecting precise and reliable DNA-based biomarkers that are missed by protein-level approaches.

Expanding to a Gram-negative pathogen, we used the CURED pipeline to identify unique sequences in *Acinetobacter baumannii* genomes belonging to the global clone 1 (GC1) compared to non-GC1 genomes (15) (Supplemental method). The GC1 clone is extensively antibiotic resistant and is one of the most disseminated high-risk clones, underscoring the urgent need for accurate detection and containment strategies (15). CURED analyzed 311 GC1 and 4,487 non-GC1 genomes, identifying 91 k-mer sequences as 100% sensitive and 100% specific to the GC1 clone, completing the analysis in just over 1 h and using 6.35 GBs of RAM (Table S3). Of these, 18 contained RE sites unique to GC1. Notably, in the study by Álvarez et al. (15), a 367-base unitig (U1) was identified with 100% sensitivity and specificity to GC1, and CURED accurately detected sequences contained within this region (Fig. S1). This finding further supports CURED's capability to accurately detect clonal signatures and highlight its robustness across both clonal (genetically homogeneous) and diverse microbial populations.

We then challenged CURED with a broader, species-wide task: differentiating toxin-positive from toxin-negative strains of *Clostridioides difficile* (Supplemental method). In *C. difficile*, the toxin genes *tcdA* and *tcdB*, which encode toxins A and B, respectively, are located within a pathogenicity locus present in toxigenic strains and absent in non-toxigenic strains (16, 17). This task differed from previous clade- or clone-based applications, as it required detecting differences across a much more diverse evolutionary backgrounds, with the goal of developing a rapid PCR test for distinguishing these two genotypes (18). In under 25 min and using less than 5 GB of RAM, CURED identified 10 k-mers that were 100% sensitive and specific to the toxigenic group, across 12,849 genomes, comprising both toxigenic and non-toxigenic strains (Table S3). Six of these sequences had unique RE sites, which were determined in less than 1 h, requiring less than 1 GB of RAM. When mapped to a toxigenic *C. difficile* genome, the k-mers were contained within the pathogenicity locus, providing independent corroboration for the significance of k-mers identified by CURED (Fig. S2). In addition to identifying the pathogenicity locus using a PCR-based detection, CURED's utility can extend to trait-level differentiation at the species level.

Finally, we tested CURED in a hospital setting of a neonatal intensive care unit (NICU). *S. aureus* commonly colonizes the skin and mucous membranes of infants and is a leading cause of invasive infections in NICUs (19, 20). Transmission of more virulent strains of *S. aureus* could increase infection risk in this vulnerable population (21). With the goal of controlling such transmission, we implemented the CURED pipeline

in the NICU environment at the Children's Hospital of Philadelphia (CHOP), leveraging the genomic data from 1,670 isolates collected during an ongoing surveillance effort (Supplementary Methods). Computational analyses were performed to characterize the composition of the circulating clones of *S. aureus* in the NICU (21). Focusing on the largest transmission cluster in the CHOP NICU, referred to as Cluster 1 by She et al. (21), we used CURED to identify three biomarkers specific to the genomes belonging to this cluster (Table S3). Following the selection of one of these biomarkers for PCR primer design, PCR assays were then developed and validated (Fig. 3A). Using a combination of the *in silico* findings along with the wet-lab protocol, the Cluster 1 membership of incoming *S. aureus*-positive samples was rapidly identified within one day—a much faster turnaround time compared to the weeks it typically takes to confirm membership with traditional sequencing methods (Fig. 3A through C ; Fig. S3). Of the 45 isolates tested using the CURED pipeline, four were positive for Cluster 1 membership. Three of these were subsequently sequenced and confirmed as Cluster 1 by WGS. Notably, none of the isolates that tested negative by CURED were later found to be Cluster 1 positive by WGS. Amplification failed in only 2 of the 45 isolates (95% amplification success rate), both of which were ultimately confirmed as negative by WGS, indicating that neither sensitivity nor specificity was affected. These failures were attributed to mutations in the primer binding sites (Supplemental method; Fig. S4). In addition to its accuracy, the CURED approach significantly reduces the cost of confirming cluster membership compared to WGS. While WGS costs approximately $70 per isolate, CURED reduces the total cost to $7.50 per isolate, including labor. Without labor costs, the reagent costs are just $0.50 per isolate. In this case, we use cluster membership to guide infection prevention and control (IPC) policy adjustments and implement targeted measures to limit transmission. As a result, this integrated approach has allowed us to focus on high-risk pathogens and inform IPC practices without relying on WGS, showcasing the power of the CURED pipeline to translate genomic insights into fast, actionable outcomes for patient safety.

Taken together, these results show that while CURED was originally developed for local investigations of emerging pathogens, it continues to perform robustly even in globally disseminated clones and, in some cases, across species-level traits (Fig. 2B). Nonetheless, broader evolutionary diversity may affect long-term marker stability, and periodic reassessment of k-mer sets may be necessary for ongoing surveillance efforts. There is also a possibility that some of the clones may have unique k-mers but do not contain a unique restriction enzyme site. In this case, effective diagnostic tests can still be designed using alternative methods such as SNP-based PCR, Sanger sequencing of diagnostic alleles, or designing k-mer-specific probes for real-time PCR. Thus, CURED both enables restriction enzyme–based diagnostics and expands the design space for complementary rapid molecular assays, depending on the user's platform and resource availability.

In conclusion, CURED has the potential to greatly improve global health monitoring by enabling equitable disease surveillance across both high- and low-resource settings, including contexts where WGS is either cost-prohibitive or logistically challenging. Designed to generate actionable insights for infection control such as identifying transmission events, environmental reservoirs, and sources of infection, CURED bridges the gap between high-resolution genomics and accessible diagnostics. Its ability to rapidly identify clonal groups across diverse pathogens, from local outbreaks to globally distributed lineages and species-wide traits, highlights its broad applicability in public health and clinical microbiology. By integrating *in silico* clone characterization with bench-level diagnostics, CURED enables earlier detection of high-risk clones and supports more agile and decentralized infection prevention responses without requiring extensive sequencing or specialized computational resources.

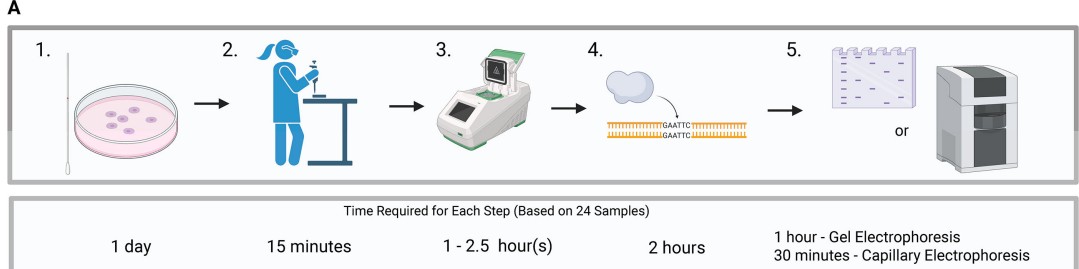

**A**

| | | | | |
|---|---|---|---|---|
| 1. | 2. | 3. | 4. | 5. |

Time Required for Each Step (Based on 24 Samples)

| 1 day | 15 minutes | 1 - 2.5 hour(s) | 2 hours | 1 hour - Gel Electrophoresis<br>30 minutes - Capillary Electrophoresis |

**Overview of CURED Protocol on Bench.** 1) *S. aureus* positive culture plates are collected from the hospital diagnostic lab after culturing from nasopharyngeal swabs. 2) PCR plates are set up using the colony PCR technique. A pipette tip is used to pick a colony from the agar plate, which is then resuspended and lysed at the bottom of the PCR tube. PCR master mix is then added to initiate the reaction. 3) A thermocycler is used to perform PCR. 4) Restriction enzyme digests for amplified products of Cluster 1. 5) Run an agarose gel or capillary electrophoresis to analyze restriction enzyme digests.

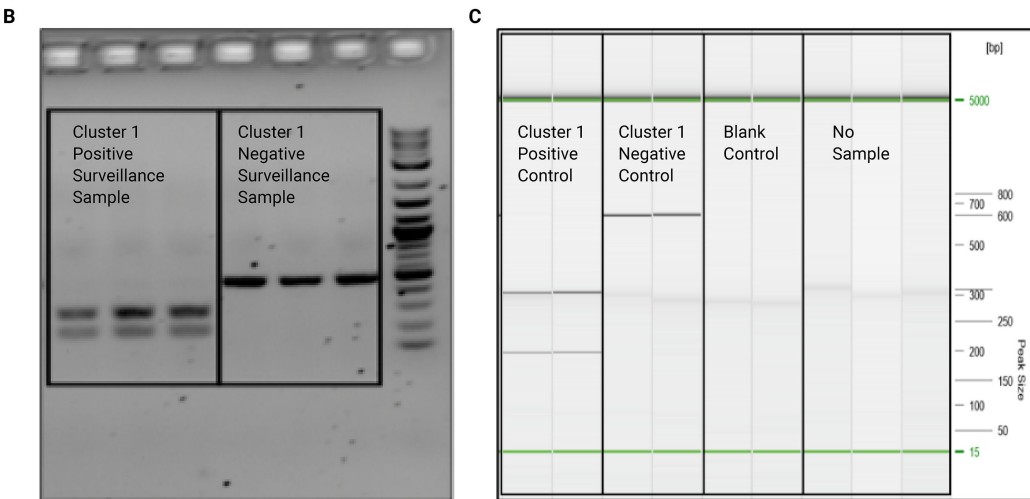

**B**

Cluster 1 Positive Surveillance Sample

Cluster 1 Negative Surveillance Sample

**C**

Cluster 1 Positive Control

Cluster 1 Negative Control

Blank Control

No Sample

[bp]

5000

800
700
600
500

300
250

200

150
100

50

Peak Size

**FIG 3** Implementation of the CURED pipeline for hospital surveillance. (A) Wet-lab workflow for applying CURED to detect *S. aureus* clusters of interest in a hospital setting. (B) Example gel electrophoresis results validating CURED Cluster 1 predictions. Lanes show Cluster 1 positive surveillance samples from the NICU and non-Cluster 1 negative control corresponding to the presence or absence of the targeted cluster-specific k-mer. (C) Example fragment analysis of the CHOP NICU samples using automated capillary electrophoresis. Uncut bands indicate absence of the biomarker k-mer, while digested bands confirm the presence of Cluster 1-associated sequences.

## MATERIALS AND METHODS

### RdJ *Staphylococcus aureus*

To download all *S. aureus* genomes from NCBI, ncbi-datasets (v 15.28.0) was used with the following command: datasets download genome taxon 'Staphylococcus aureus' --assembly-source 'GenBank' --include genome --dehydrated --filename Saureus_genomes.zip.

CURED_Main.py identified one k-mer—the RdJ k-mer—that was unique to the case group using the following command. Output files are available online to download (22). In addition to being present in 100% of the case genomes, this k-mer was also identified in 12 control genomes (Table S1). The following command was run: CURED_Main.py –case_control_file case_control_all.txt –specificity 99 –genomes_folder genomes/.

CURED_FindREs.py identified the RdJ k-mer as having three unique restriction enzyme sites, including the Bgl1 endonuclease. Output files are available online to download (22). The following command was run: CURED_FindREs.py –case_control_file case_control_all.txt –specificity 99 UniqueKmers.txt genomes/.

## USA300 *Staphylococcus aureus*

Genomes and classifications were obtained from Bianco et al. (14). In the first iteration, the 73 SAE clade genomes served as cases and the remaining genomes as controls. The following command was used to run CURED_Main.py: CURED_Main.py --genomes_folder MRSA_USA300_dataset/ --sensitivity 94 --case_control_file updated_sae.csv.

CURED_Main.py was run with a sensitivity threshold of 94%, matching the sensitivity of the diagnostic alleles identified for this clade. CURED identified 81 k-mers within this threshold. Output files are available online to download (22). Twenty of these k-mers were identified with 100% sensitivity and specificity (Table S2).

These parameters were also applied during the restriction enzyme site analysis using CURED_FindREs.py. The following command was used to run CURED_FindREs.py: CURED_FindREs.py --case_control_file ../updated_sae.csv --specificity 94 UniqueKmers.txt MRSA_USA300_dataset/.

CURED_FindREs.py identified that 39 of the 81 k-mers had at least one or more unique restriction enzyme site. Output files are available online to download (22).

For the 129 NAE-designated genomes, the following command was used to run CURED_Main.py: CURED_Main.py --genomes_folder MRSA_USA300_dataset/ --sensitivity 98 --case_control_file updated_nae.csv.

The sensitivity was set to 98% for the same reason as stated above. CURED_Main.py identified 67 k-mers unique to the NAE clade genomes within this threshold. Output files are available online to download (22). Of these 67, two were identified with 100% sensitivity and specificity (Table S2).

These parameters were also applied during restriction enzyme site analysis using CURED_FindREs.py, which identified that 17 of the 67 unique k-mers had at least one unique restriction enzyme site. Output files are available online to download (22). The following command was used to run CURED_FindRE.py for the NAE clade: CURED_FindREs.py --specificity 98 --case_control_file ../upated_nae.csv UniqueKmers.txt MRSA_USA300_dataset/.

To determine the location of the unique k-mers in the respective case genomes, blastn was used (7). For the NAE and SAE clades, the corresponding unique k-mers were queried against the representative genomes SRR5244819 and MRSA_S17, respectively: blastn -subject genome.fasta -query unique_kmers.fa -percent_identity 100 -word_size 10 -evalue 1e-2 -outfmt 6 -out blastn_results.tsv.

The coordinates of the unique k-mers were then compared to the GFF3 annotation file of the respective case genome to determine which gene the k-mer was located in, using a custom Python script.

The sensitivity and specificity of each individual k-mer identified by CURED— including any case genomes lacking the k-mer—were calculated using the raw pyseer outputs along with a custom Python script.

To determine which k-mers overlapped with diagnostic alleles for each clade, tblastn (v 2.15.0) (23) was used to identify the coordinates of diagnostic alleles within the coding sequence of SRR5244819 and MRSA_S17, the representative case genomes for the NAE and SAE clades, respectively: tblastn -query diagnostic_alleles.faa -subject genomes.fasta -outfmt 6 -out tblastn_results.tsv.

Overlaps between k-mers and diagnostic alleles were then manually compared and annotated in Table S2.

To determine the sensitivity and specificity of the dominant allele from Bianco et al. relative to the corresponding gene where CURED identified k-mers with 100% sensitivity and specificity, the following workflow was used:

i. The gene corresponding to the CURED-identified k-mers was located in the GFF3 annotation file of the aforementioned representative case genomes for both the NAE and SAE clades.

ii. The sequence of this gene was cross-referenced with Table S3 of Bianco et al. to identify the matching allele.

iii. The sensitivity and specificity values reported by Bianco et al. for this allele were then recorded.

### *Acinetobacter baumannii*

The *A. baumannii* genomes published in data set 2 from the publication by Álvarez, Verónica Elizabeth, et al., were downloaded from GenBank using ncbi-datasets with the following command (3, 15): datasets download genome accession –inputfile Ab_data-set2_accessions.txt.

Due to its suppression status on NCBI, GCA_016540885 could not be downloaded. The 311 genomes classified as being part of the GC1 clone served as the case group, while the other 4,487 non-GC1 genomes were designated as the controls (15) (Table S3). CURED_Main.py with the k-mer size set to 31 identified 91 k-mers that are unique to the GC1 clone: CURED_Main.py --genomes_folder input_genomes/ --kmer_length 31 --case_control_file Ab_case_control.csv.

Subsequently, CURED_FindREs.py evaluated the identified k-mers for potential unique restriction enzyme sites and found that 18 of the 91 had at least one unique restriction enzyme site. Output files are available online to download (22). The following command was used to run the restriction enzyme portion of CURED: CURED_FindREs.py --case_control Ab_case_control.csv UniqueKmers.txt input_genomes/.

### *Clostridioides difficile*

To identify biomarkers in toxigenic *C. difficile*, all *C. difficile* genomes available in GenBank and RefSeq at the time (February 2024) were downloaded using ncbi-datasets (3). After performing quality checks (≥95% completeness and <5% contamination) and removing assemblies with breaks in the pathogenicity locus that precluded reliable toxigenicity determination, 11,213 GenBank genomes and 1,636 RefSeq genomes were retained, yielding a total of 12,849 genomes. The combined data set was classified into 10,972 toxigenic (cases) and 1,877 non-toxigenic (controls) (18) (Table S3). Output files are available online to download (22). CURED_Main.py was used to identify 10 k-mers unique to toxigenic *C. difficile*: CURED_Main.py --number_of_cases 500 --number_of_controls 100 --case_control_file all_case_control.txt --genomes_folder ncbi_cdiff_genomes/.

Following that analysis, CURED_FindREs.py was used to identify unique restriction sites in 6 of the 10 unique k-mers: CURED_FindREs.py --case_control_file all_case_con-trol.txt UniqueKmers.txt ncbi_cdiff_genomes/.

### NICU *Staphylococcus aureus*

The genomes used in the analysis of *S. aureus* transmission within the NICU at CHOP are available as a preprint (21). Using the CHOP data set, the genomes belonging to Cluster 1 served as the case group, and the remaining genomes in this data set served as the control group. CURED_Main.py was used to find 351 k-mers unique to the genomes belonging to Cluster 1 with the following command: CURED_Main.py --genomes_folder contigs/ --case_control_file CURED_1.csv --extension fa.

As a means of filtering the number of k-mers in the search space, CURED_Main.py —with the mode set to simple—was used to query these 351 k-mers against a data set of 79,920 *S. aureus* genomes previously downloaded from GenBank in August 2024 using ncbi-datasets (Tables S1 and S3). CURED_Main.py identified that 27 of the 351 were unique to the genomes belonging to Cluster 1. Output files are available online to download (22). The following command was used: CURED_Main.py – kmer_list kmers.txt --use_simple --genomes_folder Saureus/ --case_control_file CURED_1_all.csv.

CURED_FindREs.py was used to identify potential unique restriction enzyme sites in these 27 unique k-mers using the CHOP data set. Using the following command,

CURED_FindREs.py identified that 12 of the 27 k-mers had at least one unique restriction enzyme site. Output files are available online to download (22). The following command was used: CURED_FindREs.py --extension fa --case_control_file CURED_1.csv kmer_input.txt contigs/.

Finally, CURED_FindREs.py was used to identify unique restriction enzyme sites in these 12 k-mers using the global data set of 79,920 genomes. CURED_FindREs.py identified 3 k-mers that had at least one unique restriction enzyme site. Output files are available online to download (22). The following command was used: CURED_FindREs.py --case_control_file CURED_1_all.csv Global_RE_kmers_input.txt Saureus/.

The detailed wet lab protocol is available in the Supplemental methods.

## ACKNOWLEDGMENTS

The authors would like to acknowledge the Children's Hospital of Philadelphia Microbiome Center, Infectious Diseases Diagnostic Laboratory, Neonatal Intensive Care Unit, Infection Prevention and Control, and the microbialARC strain bank for their support.

This work was funded by the following: Children's Hospital of Philadelphia (CHOP) OMICS Workstream Initiative Award (to J.P.Z., P.J.P. and A.M.M.) and support from the Center for Microbial Medicine (to J.P.Z., P.J.P. and A.M.M.); National Institutes of Health grants U19AI174998 (to A.M.M. and J.P.Z.) and 1R01AI185544-01A1 (to A.M.M, J.P.Z., P.J.P., and L.S.); Gates Foundation grant INV-065400 (to A.M.M. and S.E.C.); Conselho Nacional de Desenvolvimento Científico e Tecnológico (CNPq) grant 408725/2022-2 and Fundação Carlos Chagas Filho de Apoio à Pesquisa do Estado do Rio de Janeiro (FAPERJ) grant E-26/211.554/2021 (to A.M.S.F).

Conceptualization: A.M.S.F, P.J.P., and A.M.M. Methodology and Code development: E.T. and A.M.M. Software implementation and Data curation: E.T., A.M.M. with support from E.Q and A.D. Wet laboratory validation: S.R., S.H., D.I.T., T.N., L.H., A.S.V., M.A.C.E and A.M.S.F. Genomic analysis and Interpretation: E.T., A.F., B.E.G., A.S.V., A.M.S.F, Q.S., C.T, N.L., M.C.A., J.P.Z, P.J.P., and A.M.M. Hospital Implementation and Clinical integration: S.R., S.H., D.I.T., E.H., R.F.P., L.S., P.J.P., S.E.C., and A.M.M. Supervision and Project administration: J.P.Z., L.S., P.J.P., and A.M.M. Writing–original draft: E.T. and A.M.M. Writing–review and editing: E.T., P.J.P., and A.M.M. Funding acquisition: L.S., J.P.Z., P.J.P., and A.M.M.

## AUTHOR AFFILIATIONS

[1]Center for Microbial Medicine, Children's Hospital of Philadelphia, Philadelphia, Pennsylvania, USA

[2]Division of Gastroenterology, Hepatology, and Nutrition, Children's Hospital of Philadelphia, Philadelphia, Pennsylvania, USA

[3]Division of Pediatric Infectious Diseases, Children's Hospital of Philadelphia, Philadelphia, Pennsylvania, USA

[4]School of Engineering and Applied Science, University of Pennsylvania, Philadelphia, Pennsylvania, USA

[5]School of Arts and Sciences, University of Pennsylvania, Philadelphia, Pennsylvania, USA

[6]Department of Pediatrics, Perelman School of Medicine, University of Pennsylvania, Philadelphia, Pennsylvania, USA

[7]Departamento de Microbiologia Médica, Universidade Federal do Rio de Janeiro, Rio de Janeiro, Brazil

[8]Department of Microbiology, Perelman School of Medicine, University of Pennsylvania, Philadelphia, Pennsylvania, USA

[9]Institute for Immunology and Immune Health, Perelman School of Medicine, University of Pennsylvania, Philadelphia, Pennsylvania, USA

[10]Department of Pathology and Laboratory Medicine, Children's Hospital of Philadelphia, Philadelphia, Pennsylvania, USA

[11]Department of Pathology and Laboratory Medicine, Perelman School of Medicine, University of Pennsylvania, Philadelphia, Pennsylvania, USA

[12]Division of Protective Immunity, Children's Hospital of Philadelphia, Philadelphia, Pennsylvania, USA

[13]Division of Neonatology, Children's Hospital of Philadelphia, Philadelphia, Pennsylvania, USA

[14]Faculdade de Medicina, Programa de Pós-graduação em Patologia, Universidade Federal Fluminense, Niterói, Brazil

[15]Institute for Comparative Genomics, American Museum of Natural History, New York, New York, USA

[16]Department of Biomedical and Health Informatics, Children's Hospital of Philadelphia, Philadelphia, Pennsylvania, USA

## AUTHOR ORCIDs

Erin Theiller  http://orcid.org/0009-0001-4703-8925
Nathan L'Etoile  http://orcid.org/0009-0006-1489-1686
Michael C. Abt  http://orcid.org/0000-0002-7351-2061
Joseph P. Zackular  http://orcid.org/0000-0002-3228-3055
Paul J. Planet  http://orcid.org/0000-0003-0543-0539
Ahmed M. Moustafa  http://orcid.org/0000-0002-9949-6936

## FUNDING

| Funder | Grant(s) | Author(s) |
|---|---|---|
| National Institute of Allergy and Infectious Diseases | U19AI174998 | Joseph P. Zackular |
| | | Ahmed M. Moustafa |
| Bill and Melinda Gates Foundation | INV-065400 | Ahmed M. Moustafa |
| National Institute of Allergy and Infectious Diseases | 1R01AI185544-01A1 | Ahmed M. Moustafa |
| | | Paul J. Planet |
| | | Joseph P. Zackular |
| | | Lakshmi Srinivasan |

## AUTHOR CONTRIBUTIONS

Erin Theiller, Data curation, Formal analysis, Methodology, Software, Validation, Writing – original draft, Writing – review and editing | Swetha Rajagopol, Investigation, Validation | Stephanie Higgins, Investigation, Validation | Dayanara I. Torres, Investigation, Validation | T'Nia Napper, Investigation, Validation | Bianca E. Galis, Formal analysis | Arie Dash, Data curation, Software | Elizabeth Qian, Data curation, Software | Lauren Hamlette, Investigation | Qianxuan She, Formal analysis | Ceylan Tanes, Formal analysis | Nathan L'Etoile, Formal analysis | Andries Feder, Formal analysis | Alice Slotfeldt Viana, Formal analysis, Validation | Matheus Assis Côrtes Esteves, Validation | Michael C. Abt, Formal analysis | Susan E. Coffin, Validation | Ericka Hayes, Validation | Robert F. Potter, Validation | Joseph P. Zackular, Formal analysis, Funding acquisition, Project administration, Supervision | Lakshmi Srinivasan, Project administration, Supervision, Validation | Agnes Marie Sá Figueiredo, Conceptualization, Formal analysis, Validation | Paul J. Planet, Conceptualization, Formal analysis, Funding acquisition, Project administration, Supervision, Validation, Writing – review and editing | Ahmed M. Moustafa, Conceptualization, Formal analysis, Funding acquisition, Methodology, Project administration, Software, Supervision, Validation, Writing – original draft, Writing – review and editing

## DATA AVAILABILITY

Documentation for the usage of CURED, including explanations of all major features, is available at https://github.com/microbialARC/CURED. The data sets supporting the conclusions of this article are available on NCBI and all the accession numbers are provided in the supplemental tables. The data sets of genomes analyzed during the

current study are publicly available on GenBank with accession numbers available in Tables S1 to S3, and all output files generated in the study are available to download (22).

## ETHICS APPROVAL

Samples collected from the CHOP NICU in this study were approved by the CHOP Institutional Review Board (IRB 022889 & 17-014648).

## ADDITIONAL FILES

The following material is available online.

### Supplemental Material

**Supplemental material (mSystems01063-25-S0001.docx).** Supplemental methods and figures and captions for Table S1 to S3.
**Table S1 (mSystems01063-25-S0002.xlsx).** Genome accessions and runtime and memory metrics.
**Table S2 (mSystems01063-25-S0003.xlsx).** K-mer results from running the USA300 data set.
**Table S3 (mSystems01063-25-S0004.xlsx).** Genome accession numbers for 3 data sets.

### Open Peer Review

**PEER REVIEW HISTORY (review-history.pdf).** An accounting of the reviewer comments and feedback.

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
