## [Reviewer comments · mSystems]

Empowering Global Disease Surveillance with CURED: A Tool for Rapid Identification of Unique Genomic Biomarkers

Erin Theiller, Swetha Rajagopol, Stephanie Higgins, Dayanara Torres, T'Nia Napper, Bianca Galis, Arie Dash, Elizabeth Qian, Lauren Hamlette, Qianxuan She, Ceylan Tanes, Nathan L'Etoile, Andries Feder, Alice Slotfeldt Viana, Matheus Assis Côrtes Esteves, Michael Abt, Susan Coffin, Ericka Hayes, Robert Potter, Joseph Zackular, Lakshmi Srinivasan, Agnes Figueiredo, Paul Planet, and Ahmed Moustafa

Corresponding Author(s): Ahmed Moustafa, The Children's Hospital of Philadelphia

Review Timeline:

Submission Date:	July 17, 2025
Editorial Decision:	October 20, 2025
Revision Received:	December 18, 2025
Accepted:	January 22, 2026

Editor: Juliette Hayer

Reviewer(s): Disclosure of reviewer identity is with reference to reviewer comments included in decision letter(s). The following individuals involved in review of your submission have agreed to reveal their identity: Jonathan C Thomas (Reviewer #2)

Transaction Report:

DOI: <https://doi.org/10.1128/msystems.01063-25>

Re: mSystems01063-25 (Empowering Global Disease Surveillance with CURED: A Tool for Rapid Identification of Unique Genomic Biomarkers)

Dear Dr. Ahmed M Moustafa:

After the difficulties of the summer/September period in finding reviewers (my apologies for that) I am now happy to forward you the comments of the reviewers on your manuscript, which suggest minor modifications.

Please revise your manuscript accordingly and answer their comments and questions.

Note that Reviewer #1 could not install your tool with Conda, so please fix this installation bug so we could install and test CURED appropriately.

Revision Guidelines

Sincerely,
Juliette Hayer
Editor
mSystems

Reviewer #1 (Comments for the Author):

The authors developed a tool that can be used for identifying biomarkers specific to user-defined groups of bacteria. Although genomic data are needed for the input for the tool, users can avoid further whole-genome sequencing to detect the target of interest once the PCR protocol is developed and fixed. The findings provided in this paper is interesting, though there are concerns as below:

Why didn't the authors suggest SNP-based PCR assay (i.e., PCR targeting SNPs specific to the case group)? (e.g., doi: 10.1128/AAC.01604-13) As the authors indicated in figure 2C, restriction enzyme digesting requires a few hours.

Line 117 "biomarker": Please explain the relationship/difference between "biomarker" and "unique k-mer"/"unique restriction enzyme site". Is the definition used for biomarker the same throughout the manuscript? For example, it seems biomarkers are used for unique k-mers in Figure 1 and for unique k-mers plus unique restriction enzyme sites in the title of this paper. If this is the case, please be consistent.

Lines 134-138: What happens if the ST(s) originating from the target ST is present in the database? If this ST is present in the dataset, does CURED return that there is no specific marker for the target ST under the default setting (i.e., sensitivity of 100%, specificity of 100%)? Do the users have to manually screen these STs in the control group, include these in the target group, and redo the analysis again? It would be helpful if CURED has an option to specify ST complex as a target group.

Lines 187-188: What is the difference between "restriction enzyme site" and "RE site"?

Lines 249-250: Is 17 hours short enough to enable "real-time" genomic surveillance?

Line 308: What does CHOP mean?

Lines 444-448: Please briefly explain the reason why $19,645 + 1,636$ is not equal to $10,972 + 1,877$.

Throughout the manuscript: Please indicate the version for each software (e.g., bwa-mem and blastn).

Figure 2C: Please use a high-resolution figure for this.

Figure 2D and 2E: What is the difference between these two figures? Are these figures redundant?

Figures 2A and 2B are related to in silico analysis, and figures 2C-2E are related to laboratory experiments. At least these figures should be separated (i.e., figures 2C, 2D, and 2E as figures 3A, 3B, and 3C).

I tried to install the software using conda:

```
conda activate env_cured
conda install bioconda::cured
```

However, it failed as below:

```
Platform: osx-arm64
Collecting package metadata (repodata.json): done
Solving environment: failed
```

```
LibMambaUnsatisfiableError: Encountered problems while solving:
- nothing provides blast 2.15.0 needed by cured-1.05-hdfd78af_0
```

```
Could not solve for environment specs
The following package could not be installed
└─ cured is not installable because it requires
└─ blast 2.15.0 , which does not exist (perhaps a missing channel).
```

Reviewer #2 (Comments for the Author):

The authors present a new bioinformatics tool, CURED, designed to identify unique genomic regions specific to a clone or pathogenic variant of a species, that might be amplified by PCR and digested with restriction enzymes. This will allow the identification of said clones without the need for whole-genome sequencing.

The manuscript is clearly written in easy to understand English, and lays out how CURED works, benchmarking against other similar tools and its results for five previously analysed datasets.

Some minor comments:

Line 191: "Once identified, the default number of bases, which is 20, is added to either end of the sequence, elongating the sequence to 60 bases in length." How does CURED handle consecutive kmers unique to the case group. Are they merged into one unique region? Otherwise neither of the flanks of the elongated sequence might be found in the control group?

Line 234: "in a runtime comparable to KEC". It was comparable for 20k, 40k and 80k genomes. But for smaller datasets at 100 genomes, KEC is 4-5x faster, and at 1000 genomes, 3x faster. Could be a bit more specific regarding this point here.

Line 356: "To download all *S. aureus* genomes from NCBI, the following command was used". I initially read this as being a CURED command, but on line 430, the command to download the *A. baumannii* genomes has a similar structure and there you state this is a ncbi-datasets command (also, all other CURED commands start CURED_Main.py or CURED_FindRes.py). Could you just make it clear which program is being used on line 356 to download genomes?

Line 365: should that be 'as' rather than 'has'?

Supplementary material:

Wet lab protocol - primer design. Did primers have to be within a set distance of the kmer? Or was this varied from kmer to kmer, to give interpretable band sizes post digest?

"the efficiency of the restriction enzyme associated with the kmer". How was this determined?

Methods of figures 2. Panel A - could give the machine specs here?

One final thing I'm curious about (and I am in no way suggesting that the authors perform this test for the manuscript, it's only my own curiosity) - have they looked at all at how CURED might perform for two or three very closely related species, compared to an outgroup? Or would that not be feasible?

Dear Reviewers and Editors of ASM mSystems,

Thank you for your thorough and thoughtful review of our manuscript, "Empowering Global Disease Surveillance with CURED: A Tool for Rapid Identification of Unique Genomic Biomarkers" (mSystems01063-25). Below, we have responded to the reviewer's comments, and we believe that our manuscript is improved based on their questions and recommendations. We have included our narrative response in blue and additions/changes to the manuscript in red.

Reviewer #1 (Comments for the Author):

The authors developed a tool that can be used for identifying biomarkers specific to user-defined groups of bacteria. Although genomic data are needed for the input for the tool, users can avoid further whole-genome sequencing to detect the target of interest once the PCR protocol is developed and fixed.

We thank the reviewer for this accurate and concise summary of the tool's purpose and utility and for their time to review our manuscript.

The findings provided in this paper is interesting, though there are concerns as below:

Why didn't the authors suggest SNP-based PCR assay (i.e., PCR targeting SNPs specific to the case group)? (e.g., doi: 10.1128/AAC.01604-13) As the authors indicated in figure 2C, restriction enzyme digesting requires a few hours.

We appreciate the reviewer's thoughtful suggestion. SNP-based PCR assays are indeed a well-established approach for molecular typing and variant detection. Our decision to prioritize a restriction enzyme-based strategy in CURED was motivated primarily by accessibility, robustness, and interpretability in resource-limited settings, which represent a major intended application for this tool.

First, CURED was intentionally designed to rely on widely available laboratory infrastructure: standard PCR, commercially available restriction enzymes, and conventional gel electrophoresis (or basic fragment analysis). In many low- and middle-income country (LMIC) laboratories where high-resolution real-time PCR instruments and allele-specific probes are not consistently available, this approach offers a significantly lower barrier to implementation while maintaining strong discriminatory power.

Second, in many instances the CURED design allows for successful amplification in both case and control isolates, with differentiation achieved by a distinct restriction digestion pattern. This provides an added layer of interpretability, as successful amplification confirms assay performance before classification. We acknowledge that, in some cases, the target locus may be

truly specific to the case group and therefore absent in controls, resulting in a presence/absence pattern similar to SNP-based assays. However, when a locus is selected to be shared across the two groups and contains a lineage-specific restriction site, the digestion-based strategy reduces ambiguity between a true negative and a failed PCR reaction which is an important practical advantage in real-world laboratory settings.

Third, while the restriction digestion step adds modest time (typically 1–2 hours), the overall workflow still delivers same-day, actionable results and remains substantially faster and more accessible than sequencing-based confirmation. In our implementation, this trade-off was considered acceptable in exchange for broader durability, robustness to sequence variation, and greater global applicability.

Importantly, CURED does not exclude SNP-based methodologies. Once lineage-specific k-mers are identified, they can also be used to inform SNP-based or probe-based designs in settings where such platforms are available. In this sense, CURED is intended to complement rather than replace SNP-based PCR, while specifically expanding access to genomic-informed diagnostics in settings with limited infrastructure. For added clarity, we have revised the text as follows:

Lines 379-384: “There is also a possibility that some of the clones may have unique k-mers but do not contain a unique restriction enzyme site. In this case, effective diagnostic tests can still be designed using alternative methods such as SNP-based PCR, Sanger sequencing of diagnostic alleles or designing k-mer-specific probes for real time PCR. Thus, CURED both enables restriction enzyme–based diagnostics and expands the design space for complementary rapid molecular assays, depending on the user’s platform and resource availability.”

Line 117 "biomarker": Please explain the relationship/difference between "biomarker" and "unique k-mer"/"unique restriction enzyme site". Is the definition used for biomarker the same throughout the manuscript? For example, it seems biomarkers are used for unique k-mers in Figure 1 and for unique k-mers plus unique restriction enzyme sites in the title of this paper. If this is the case, please be consistent.

We thank the reviewer for highlighting this point. We use the term biomarker to refer to a unique k-mer that has a unique restriction enzyme site. We have reviewed the manuscript to ensure that all uses of biomarker are consistent with this definition. For added clarity, we have revised the text as follows:

Line 118-120: “CURED_FindREs.py is run in the second part of the pipeline to scan these k-mer for k-mers with unique restriction enzyme sites that are present only in the case group, which are referred to as biomarkers in the context of this study.”

Lines 134-138: What happens if the ST(s) originating from the target ST is present in the database? If this ST is present in the dataset, does CURED return that there is no specific marker

for the target ST under the default setting (i.e., sensitivity of 100%, specificity of 100%)? Do the users have to manually screen these STs in the control group, include these in the target group, and redo the analysis again? It would be helpful if CURED has an option to specify ST complex as a target group.

We thank the reviewer for this thoughtful and important question. If the user selects the ST + species option, CURED performs MLST during genome collection. In this scenario, if genomes belonging to the specified target ST are already present in the database, they are automatically reassigned to the case group prior to marker discovery. This preserves the default requirement of 100% sensitivity and 100% specificity and prevents a false “no marker found” outcome.

However, when users provide custom genome accessions or FASTA files to define case and control groups, CURED intentionally does not perform MLST, as group definitions in many real-world applications are not based on sequence type alone. For example, in outbreak investigations and genomic surveillance, analyses are often conducted within a single ST or clonal complex to distinguish a specific outbreak clone or sub-lineage from its very closely related background strains. In such cases, classifications may be based on epidemiological links, phenotypic traits, virulence, or outbreak association, rather than ST.

This is illustrated by our prior work on the Rio de Janeiro (RdJ) MRSA clone (ST105/CC5), where the objective was to identify strain-specific biomarkers within an otherwise highly homogeneous ST background. In that study, whole-genome analysis was required to distinguish RdJ from other closely related CC5 strains due to their high genetic similarity (references 12 and 13 in the main text). Automatically reassigning genomes purely based on ST in such contexts would be inappropriate, as it would eliminate the very signal the analysis aims to capture.

Therefore, when user-supplied genomes are provided, any reassignment based on ST would require manual intervention, which is currently not implemented by design. This preserves flexibility for phenotype-based, epidemiology-based, or sub-ST analyses. Implementation of optional automated reassignment for these cases is technically feasible and may be added in future releases as a user-selectable feature.

At this time, CURED also does not support defining clonal complexes (e.g., CCs) as target groups, since this information is not consistently available in a standardized, downloadable format from current MLST repositories. Future updates may introduce a user-defined grouping feature for public genomes (including clonal complexes). For now, we focused on ST designations because they are broadly available and consistently assigned across diverse species. However, the rapid growth of large-scale bacterial genome initiatives, such as the All the Bacteria (ATB) project (>2.4 million assemblies) and emerging platforms that integrate genome sequences with metadata (e.g., ST, predicted virulence, and AMR profiles), will further expand CURED’s reach and flexibility for both surveillance and diagnostic biomarker discovery. We have now added a clarifying text for this point.

Lines 138-141: “In contrast, when users provide their own case and control genomes based on non-ST criteria (e.g., phenotype, outbreak association, or clinical outcome) as in entry point one, MLST is not performed, and group assignment is fully user-defined.”

Lines 187-188: What is the difference between "restriction enzyme site" and "RE site"?

We thank the reviewer for the opportunity to clarify this point. There is no difference between restriction enzyme site and RE site. RE is abbreviated for “restriction enzyme.” We have defined it at first use.

Lines 103-104: “Here, we introduce the Classification Using Restriction Enzymes Diagnostics (CURED) pipeline, which uses local and public sequencing data, integrating rapid k-mer analysis and restriction enzyme (RE)-based detection in a single pipeline.”

Lines 249-250: Is 17 hours short enough to enable "real-time" genomic surveillance?

We thank the reviewer for this important question. The ~17-hour runtime corresponds specifically to a one-time restriction enzyme validation step performed during assay development, in which candidate k-mers are screened against a very large global control dataset (>79,000 genomes) to ensure maximal specificity and eliminate false positives. This step is intentionally stringent but not required for routine surveillance once a biomarker is identified.

After a unique k-mer and restriction enzyme combination has been established, the resulting assay can be deployed as a PCR- and restriction enzyme–based diagnostic, enabling same-day detection of the target lineage without the need for additional sequencing. From an infection prevention and control standpoint, it is the speed of actionable detection during routine implementation, not the one-time computational design step, that defines real-time surveillance.

We have clarified this distinction in the revised manuscript in lines 270-274.

Lines 272-276: “Although the restriction enzyme validation phase required roughly 17 hours when applied to a very large global dataset (>79,000 genomes), this step is performed only once per target lineage during assay development. Once a unique k-mer-restriction enzyme combination has been identified, the resulting assay can be deployed as a PCR diagnostic test, enabling same-day detection without the need for further sequencing.”

Line 308: What does CHOP mean?

We thank the reviewer for pointing out the need for clarification. CHOP is an abbreviation for Children’s Hospital of Philadelphia. We have defined it at first use.

Line 339: “With the goal of controlling such transmission, we implemented the CURED pipeline in the NICU environment at Children’s Hospital of Philadelphia (CHOP) leveraging the genomic

data from 1,670 isolates collected during an ongoing surveillance effort (Supplementary Methods).”

Lines 444-448: Please briefly explain the reason why $19,645 + 1,636$ is not equal to $10,972 + 1,877$.

We appreciate the reviewer’s careful attention to this methodological detail. After downloading the 19,645 GenBank genomes, we used CheckM to assess genome quality. Genomes with $<95\%$ completeness or $\geq 5\%$ contamination were excluded. Additionally, only genomes with an available corresponding amino acid FASTA file in NCBI were retained, resulting in 14,186 genomes. When determining toxigenicity status, another 2,973 genomes were excluded due to assembly breaks in the Pathogenicity Locus region that prevented toxigenicity classification. Consequently, 11,213 GenBank genomes remained. All 1,636 RefSeq genomes met the same quality criteria and were retained, yielding a total of 12,849 genomes used in the analysis. This workflow is described in the referenced publication and has been clarified in the revised manuscript.

Lines 493-499: “To identify biomarkers in toxigenic *C. difficile*, all *C. difficile* genomes available in GenBank and RefSeq at the time (February 2024) were downloaded using ncbi-datasets. (3). After performing quality checks ($\geq 95\%$ completeness and $< 5\%$ contamination) and removal of assemblies with breaks in the Pathogenicity Locus that precluded reliable toxigenicity determination, 11,213 GenBank genomes and 1,636 RefSeq genomes were retained, yielding a total of 12,849 genomes. The combined dataset was classified into 10,972 toxigenic (cases) and 1,877 non-toxigenic (controls) (18) (Supplementary Table 3).”

Throughout the manuscript: Please indicate the version for each software (e.g., bwa-mem and blastn).

We thank the reviewer for this helpful suggestion. These version numbers have been added to the revised manuscript in the Implementation section under Results and Discussion as well as the Methods and Materials section.

Figure 2C: Please use a high-resolution figure for this.

We appreciate the reviewer’s comment regarding figure clarity. This has been added to the revised manuscript in Figure 3A.

Figure 2D and 2E: What is the difference between these two figures? Are these figures redundant?

We thank the reviewer for raising this point. Figure 2D shows the confirmation of a Cluster 1 *S. aureus* sample using agarose gel electrophoresis, while Figure 2E shows the same confirmation using capillary electrophoresis. Although both methods produce the same result, we included both to demonstrate that restriction enzyme digest analysis in the wet-lab implementation of CURED can be performed using different approaches, each with distinct advantages. In particular, capillary electrophoresis provides an automated, higher-resolution alternative to agarose gel electrophoresis, enabling more standardized and efficient analysis.

Figures 2A and 2B are related to in silico analysis, and figures 2C-2E are related to laboratory experiments. At least these figures should be separated (i.e., figures 2C, 2D, and 2E as figures 3A, 3B, and 3C).

Thank you for this suggestion. The in-silico performance comparisons are now figures 2A and 2B, and the laboratory-related panels are now figures 3A-3C in the revised manuscript.

I tried to install the software using conda:

```
conda activate env_cured
conda install bioconda::cured
```

However, it failed as below:

```
Platform: osx-arm64
Collecting package metadata (repodata.json): done
Solving environment: failed
```

```
LibMambaUnsatisfiableError: Encountered problems while solving:
- nothing provides blast 2.15.0 needed by cured-1.05-hdfd78af_0
```

Could not solve for environment specs

The following package could not be installed

- └─ cured is not installable because it requires
- └─ blast 2.15.0 , which does not exist (perhaps a missing channel).

Thank you for bringing this to our attention. The reported error occurs on systems using the osx-arm64 architecture (e.g., Apple Silicon M1/M2 machines) due to a current compatibility issue with the BLAST (v2.15.0) dependency in conda on that platform. We were not able to replicate this installation error on our end using x86-based Linux and macOS systems, where CURED installs successfully via Bioconda. As Conda environments and available binaries can differ across platforms, we now recommend that users encountering this issue either (i) follow the manual installation instructions provided in the repository or (ii) use the newly provided Docker image, which contains all required dependencies preconfigured and enables platform-independent installation and execution. Instructions for both options have now been added to the

Supplementary Materials and on the CURED GitHub page to improve reproducibility and ease of use for future users.

Reviewer #2 (Comments for the Author):

The authors present a new bioinformatics tool, CURED, designed to identify unique genomic regions specific to a clone or pathogenic variant of a species, that might be amplified by PCR and digested with restriction enzymes. This will allow the identification of said clones without the need for whole-genome sequencing.

The manuscript is clearly written in easy to understand English, and lays out how CURED works, benchmarking against other similar tools and its results for five previously analysed datasets.

We appreciate the reviewer's recognition of CURED's utility and the clear presentation of our benchmarking results.

Some minor comments:

Line 191: "Once identified, the default number of bases, which is 20, is added to either end of the sequence, elongating the sequence to 60 bases in length." How does CURED handle consecutive kmers unique to the case group. Are they merged into one unique region? Otherwise neither of the flanks of the elongated sequence might be found in the control group?

Under the hood, CURED uses unitig-caller to identify k-mers that are unique to the case group. Unitig-caller extends k-mers into the longest continuous unique sequence, rather than treating each k-mer independently. Therefore, if consecutive k-mers are unique to the case group, they are merged into a single contiguous unique region. This approach ensures that the flanking regions of the elongated sequence are properly represented and reduces the chance that neither flank would be found in the control group. We have now added a clarification to this point.

Lines 169-176: "To further increase the efficiency of the tool different modes of unitig-caller are applied. The default mode is "call" mode, in which unitigs are extracted by constructing a population graph and returning the colors associated with the unitigs. In this mode, unitig-caller also extends k-mers into the longest continuous unique sequence. As a result, consecutive k-mers that are unique to the case group are merged into a single contiguous region. This ensures that the flanking regions of the extended sequence are properly represented and reduces the likelihood that either flank would be absent from the control group."

Line 234: "in a runtime comparable to KEC". It was comparable for 20k, 40k and 80k genomes. But for smaller datasets at 100 genomes, KEC is 4-5x faster, and at 1000 genomes, 3x faster. Could be a bit more specific regarding this point here.

Thank you for the thorough review. We have edited the sentence to include the specific datasets for which CURED and KEC have comparable runtimes in the revised manuscript for added clarity.

Lines 253-255: “With default parameters, CURED achieved biomarker identification in a runtime comparable to KEC for the datasets with 10,000 genomes or more, but with more than an order of magnitude lower memory requirement—less than 3 GB of RAM versus over 135 GB for KEC.”

Line 356: "To download all *S. aureus* genomes from NCBI, the following command was used". I initially read this as being a CURED command, but on line 430, the command to download the *A. baumannii* genomes has a similar structure and there you state this is a ncbi-datasets command (also, all other CURED commands start CURED_Main.py or CURED_FindRes.py). Could you just make it clear which program is being used on line 356 to download genomes?

We thank the reviewer for this helpful comment. The ncbi-datasets tool was used to download all *S. aureus* genome from NCBI. This has been added in the revised manuscript.

Line 400: “To download all *S. aureus* genomes from NCBI, ncbi-datasets was used with the following command:”

Line 365: should that be 'as' rather than 'has'?

We thank the reviewer for catching this typo. This has been corrected to “as” in the revised manuscript.

Line 411: “CURED_FindREs.py identified the RdJ k-mer as having three unique restriction enzyme sites, including the BglI endonuclease.”

Supplementary material:

Wet lab protocol - primer design. Did primers have to be within a set distance of the kmer? Or was this varied from kmer to kmer, to give interpretable band sizes post digest?

We appreciate the reviewer’s careful reading of the manuscript. For primer design, the software was configured to select primers located at least 100 bp upstream and downstream of the target k-mer to ensure the full k-mer region was captured within the amplicon. The exact distance was not fixed; in practice, primers were positioned approximately 100–250 bp (or more) from the k-mer, depending on local sequence constraints and primer quality metrics. Final primer pairs were selected based on the expected total amplicon length and the predicted sizes of the digested and undigested fragments, such that products could be clearly resolved on a 1.5% agarose gel. Importantly, primer placement is flexible and user-defined; the strategy described here reflects

the design constraints used in this study. We have also now revised this section in the supplementary material.

Primer Design Section: “We designed primers for each k-mer for Cluster 1 using the Primer3web 4.1.0 software (2), positioning primers a minimum of 100 bp upstream and downstream of the k-mer, with final distances determined by local sequence composition and primer quality metrics. The k-mers were then ranked based on primer melting temperature, GC content, the efficiency of the restriction enzyme associated with the k-mer (as inferred from recognition sequence length, with preference for longer cutters such as 6-base recognition sites), and the sizes of the pre- and post-digested products, using the SnapGene 8.0.3 software (3), to ensure clear separation on a 1.5% agarose gel.”

"the efficiency of the restriction enzyme associated with the kmer". How was this determined?

We appreciate the opportunity to clarify this point. In this context, “efficiency” refers to the restriction enzyme’s level of specificity, as determined primarily by the length of its recognition sequence. Enzymes recognizing longer motifs (e.g., 6-base cutters) are less likely to cut at random genomic locations than 4- or 5-base cutters and were therefore considered more specific and more suitable for distinguishing the target k-mer-containing region. Because multiple candidate k-mers and associated restriction enzymes were available, each enzyme was evaluated based on its recognition sequence length, and preference was given to longer cutters when other factors were comparable. We have also now revised this section in the supplementary material as shown in the previous comment.

Methods of figures 2. Panel A - could give the machine specs here?

We thank the reviewer for the comment. This has been added to the revised manuscript.

Methods of Figure 2 section: “All benchmarking was performed on a Linux machine with access to 32 CPU cores and 140 GB of RAM. To ensure consistent performance comparisons across tools, analyses were restricted to a single CPU core.”

One final thing I'm curious about (and I am in no way suggesting that the authors perform this test for the manuscript, it's only my own curiosity) - have they looked at all at how CURED might perform for two or three very closely related species, compared to an outgroup? Or would that not be feasible?

Thank you for this thoughtful question. Although not included in the manuscript, we did conduct a preliminary analysis using CURED to identify potential universal biomarkers specific to *Clostridioides difficile* that could be used for rapid species-level identification. While this analysis does not directly address your question regarding closely related species, it demonstrates a conceptually similar application. For this analysis, we downloaded all RefSeq genomes from

NCBI via ncbi-datasets, designating *C. difficile* genomes as the target group (n = 2,506) and all other bacterial genomes as the control group (n = 308,377). Running CURED_Main.py identified 31 k-mers unique to *C. difficile*.

Due to time constraints, we did not run CURED_FindRE.py, but primers were designed based on one of the unique k-mers and tested on *C. difficile* strains NTCD-035 (non-toxigenic, in-house isolate) and CD196, as well as *Enterococcus gallinarum* and two unknown isolates (later confirmed as CD196 by ribotyping). Amplification occurred only for *C. difficile* samples, demonstrating species specificity. In parallel, we have been contacted recently by a group who is leveraging CURED to detect specific taxa of interest from metagenomic datasets, suggesting broader applicability of the framework. A systematic evaluation of CURED for resolving multiple closely related species against appropriate outgroups represents an important future direction and will be explored in subsequent work.

Re: mSystems01063-25R1 (Empowering Global Disease Surveillance with CURED: A Tool for Rapid Identification of Unique Genomic Biomarkers)

Dear Dr. Ahmed M Moustafa:

The reviewers and I are satisfied by the revision of the manuscript and by your answers; all their comments have been addressed.

I underline that packaging the code into a container was a good option and now it can be run from any platform.

Just a last comment regarding the ASM's data policy:

The authors could add a Data Availability paragraph in which they provide the url of the repository and the accession numbers of the sequencing datasets used (e.g. for the NICU *S. aureus* project).

The information will therefore be more straight-forward to find.

Your manuscript has been accepted, and I am forwarding it to the ASM production staff for publication. Your paper will first be checked to make sure all elements meet the technical requirements. ASM staff will contact you if anything needs to be revised before copyediting and production can begin. Otherwise, you will be notified when your proofs are ready to be viewed.

Sincerely,
Juliette Hayer
Editor
mSystems

Reviewer #1 (Comments for the Author):

The authors addressed my concerns.

Reviewer #2 (Comments for the Author):

The authors have thoroughly addressed all points raised in the previous review.